# CALIBRATION BOTTLENECK: WHAT MAKES NEURAL NETWORKS LESS CALIBRATABLE?

## ABSTRACT

While modern deep neural networks have achieved remarkable success, they have exhibited a notable deficiency in reliably estimating uncertainty. Many existing studies address the uncertainty calibration problem by incorporating regularization techniques to penalize the overconfident outputs during training. In this study, we shift the focus from the miscalibration encountered in the training phase to an investigation of the concept of *calibratability*, assessing how amenable a model is to be recalibrated in post-training phase. We find that the use of regularization techniques might compromise calibratability, subsequently leading to a decline in final calibration performance after recalibration. To identify the underlying causes leading to poor calibratability, we delve into the calibration of intermediate features across neural networks' hidden layers. Our study demonstrates that the overtraining of the top layers in neural networks poses a significant obstacle to calibration, while these layers typically offer minimal improvement to the discriminability of features. Based on this observation, we introduce a *weak classifier hypothesis*: Given a weak classification head, the bottom layers of a neural network can be learned better for producing calibratable features. Consequently, we propose a progressively layer-peeled training (PLT) method to exploit this hypothesis, thereby enhancing model calibratability. Our comparative experiments show the effectiveness of our method, which improves model calibration and also yields competitive predictive performance.

## 1 INTRODUCTION

For a machine learning system, reliable predictive models should not only yield high accuracy but also offer heightened uncertainty when their predictions are prone to inaccuracy. While modern deep neural networks have achieved remarkable success in high-dimensional prediction tasks like computer vision, speech recognition, and natural language processing, they have exhibited a notable deficiency in reliably estimating uncertainty (Guo et al., 2017; Minderer et al., 2021). This uncertainty issue can lead to adverse consequences, particularly in scenarios involving safety-critical decision-making, and therefore, it has been the subject of extensive research in recent years (Gupta et al., 2021; Ashukha et al., 2019; Wang et al., 2021; 2023). To systematically investigate DNNs' uncertainty estimation problem, Guo et al. (2017) conducted a comprehensive study within this context and made two significant observations: (i) The miscalibration of model confidence, which can be used to reflect the uncertainty degree, is closely associated with large model capacity and the absence of regularization in training. And (ii) simple post-hoc methods like temperature scaling (Platt et al., 1999) and histogram binning (Zadrozny & Elkan, 2001) can effectively address the miscalibration issue.

Taking inspiration from the study of (Guo et al., 2017), two predominant strategies have been extensively investigated to improve the calibration performance: the first involves incorporating regularization during training, while the second entails recalibrating models in post-training stage. For the former strategy, some well-known methods which initially designed to improve generalization have been found also beneficial for calibration, including label smoothing (LS) (Müller et al., 2019), weight decay (WD) (Guo et al., 2017), mixup (MT) (Thulasidasan et al., 2019), self-distillation (SD) (Guo et al., 2021) and focal loss (Mukhoti et al., 2020). The latter strategy aims to recalibrate predictions in a post-hoc fashion by establishing a mapping from raw outputs to well-calibrated confidences. Typically, these approaches usually incorporate extra parameters which necessitate tuning

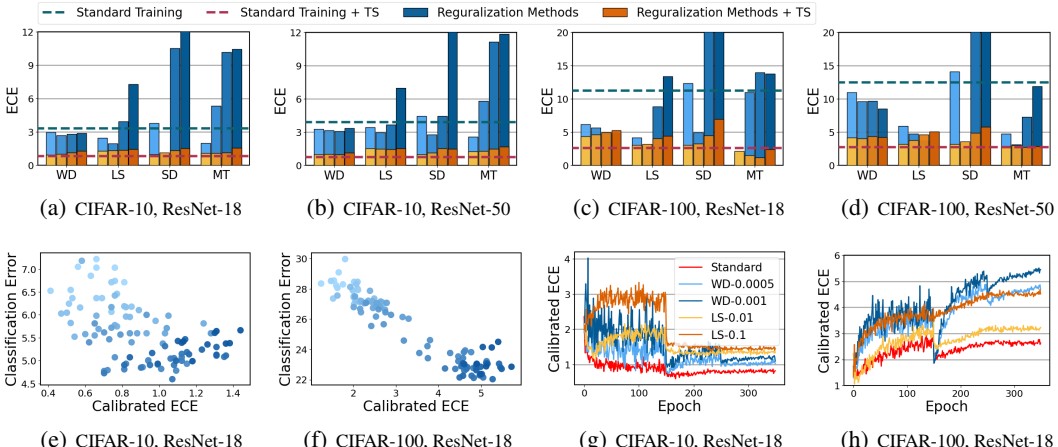

Figure 1: (a-d): ECE of regularization methods. We conduct experiments using these regularization methods with the following coefficient sets: $\{2.5 \times 10^{-4}, 5 \times 10^{-4}, 7.5 \times 10^{-4}, 1 \times 10^{-3}\}$ for weight decay, $\{0.005, 0.01, 0.05, 0.1\}$ for Label smoothing, $\{0.1, 0.5, 1.0, 5.0\}$ for Mixup, $\{0, 1, 2, 3, 4\}$ for distillation. The role of these coefficients can be found in Appendix A.1 (e-f): Classification error and ECE of a bunch of models trained with varied weight decay coefficients. Darker points represent models trained with stronger weight decay. (g-h): The calibrated ECE over training dynamics.

on further validation data during the post-hoc calibration. There is a surge of studies aimed at improving calibration by designing new post-hoc calibration approaches (Müller et al., 2019; Mukhoti et al., 2020; Joo & Chung, 2021; Kull et al., 2019; Rahimi et al., 2020; Gupta et al., 2021). Although numerous post-hoc calibration approaches have been proposed in recent years, temperature scaling remains the most widely used one due to its simplicity and effectiveness (it only needs one parameter, i.e., the temperature in softmax layer, to be adjusted in post-hoc calibration phase).

As the aforementioned two strategies can individually improve the calibration performance at different phases, it is natural to consider their integration for superior results. However, previous studies found that well calibrated models, which are usually trained with regularization techniques, do not necessarily achieve better calibration performance after post-hoc calibration phase than models from standard training (Ashukha et al., 2019; Wang et al., 2021; Zhu et al., 2023; Wang et al., 2023; Tao et al., 2023). Remarkably, several studies have identified a counterintuitive relationship between the calibration performance before and after post-hoc calibration when the models are trained with various regularization techniques (Wang et al., 2021; Zhu et al., 2023; Wang et al., 2023). Specifically, while these regularization methods prove advantageous in boosting the calibration performance in training phase, they tend to adversely make models less calibratable, thereby compromising the final calibration performance after post-hoc calibration. As the study in (Ashukha et al., 2019) suggested, comparisons between different models might be inequitable if the post-hoc calibration is not taken into consideration. Therefore, in this study, we shift the focus from the calibration challenges encountered in training phase to an investigation of the concept of calibratability, which assesses a model's amenability to be well recalibrated in post-training phase.

**Calibration vs calibratability.** As is shown in Figures 1(a) to 1(d), we start with an experiment involving several regularization techniques, leading to the following observations that highlight the role of calibratability: (1) Without post-hoc calibration, the regularization techniques do help calibration compared with standard training. However, their calibration performance varies substantially across different regularization strengths. The calibration results appear to follow a V-shaped trend, indicating the need for careful hyperparameter tuning to achieve optimal calibration. (2) Regardless of the regularization techniques employed during training, calibration error yields significant reductions after post-hoc calibration, and the calibrated ECE is largely insensitive to hyperparameter choices. (3) Our experimental results corroborate previous studies that employing regularization techniques may inadvertently reduce a model's final calibration performance after post-hoc calibration (Wang et al., 2021; Zhu et al., 2023; Wang et al., 2023).

**The trade-off between calibratability and accuracy.** In the above experiment, we also observe a trade-off between calibratability and accuracy. As shown in Figures 1(e) and 1(f), we plot the calibrated ECE and accuracy of a bunch of models trained with different weight decay coefficients.

It is shown a negative correlation between these two metrics with Pearson correlation coefficient as -0.52 and -0.95 on CIFAR-10 and CIFAR-100, respectively. This observation raises a particularly thorny question: Are calibratability and discriminability inherently at odds within neural networks? Rather than delving directly into this question, in Section 3, we introduce a method that can improve calibration and classification performance simultaneously.

**Identifying the causes leading to poor calibratability.** As illustrated in Figures 1(g) and 1(h), we also observe a phenomenon that calibrated ECE worsens over the course of training epochs, signaling a decline in calibratability during learning dynamics. Notably, the utilization of regularization techniques can result in a significant raise in calibrated ECE after a certain number of epochs. Since the decline in calibratability and the improvement in predictive performance occur simultaneously during training, this again raises the previously mentioned concern about the conflict between the two. To investigate the reasons for the decline in neural networks' calibratability, we delve into the calibratability of the intermediate features of the hidden layers of neural networks. We find that the calibratability of these feature across layers from the bottom to the top exhibits a trend of initial decline followed by a subsequent increase. We find that this phenomenon can be connected with the information bottleneck principle in deep learning, which induces an unexpected negative impact of the information compression for calibration. Based on the empirical study, we identify the specific reason of the decline of calibratability is the overtraining of a few top layers in deep neural networks, and introduce a weak classifier hypothesis for calibration: learning calibratable features for neural networks that requires a weaker classification head.

Consequently, we propose an efficient and effective training strategy called progressively layer-peeled training (PLT) to overcome the calibration bottleneck and hence improve models' calibratability. Our method gradually freeze the parameters of top layers during the training process, to ensure that the top layers of the neural network do not excessively compress information, thereby enhancing the model's calibratability. We carry out comparative experiments on multiple datasets with different model architectures. We use scaling-based post-hoc calibration approaches to recalibrate the neural networks and evaluate both calibration and classification performance. The empirical results show that our method enhances calibratability without sacrificing predictive accuracy, which makes the concern about the inherent tension between discriminability and calibratability can be eliminated to some extent.

**Our contributions** can mainly be summarized as three-fold:

- We study the interaction of train techniques and post-hoc calibration by introducing a new property named calibratability. We observed that most train-time calibration methods struggle to learn a model with good calibratability.
- We delve into the calibratability of intermediate features across the hidden layers and find that the calibratability of these feature from the bottom to the top exhibits a U-shaped trend with initial decline followed by a subsequent increase.
- We empirically identify the reason of the decline of calibratability is the overtraining of a few top layers. In particular, we introduce a weak classifier hypothesis, namely, learning calibratable features for neural networks that requires a weaker classifier. Consequently, an simple yet effective method called PLT is proposed to improve models' calibratability. We validate our method with a series of experiments and analyses.

## 2 BACKGROUND

### 2.1 EVALUATION METRICS

**Expected Calibration Error (ECE).** ECE (Naeini et al., 2015) is a commonly used metric to measure the calibration performance. It works by firstly binning $n$ test samples into $M$ bins $\{B_m\}_{m=1}^{M}$ with respect to their confidence scores, then calculating the average difference between the accuracy and average confidence over all bins: $\text{ECE} = \sum_{m=1}^{M} \frac{|B_m|}{n} |\text{acc}(B_m) - \text{avgConf}(B_m)|$. There are two strategy for binning samples: binning with equally confidence interval and adaptive binning with equal sample mass.

**Negative Log Likelihood (NLL).** NLL will be minimized when model outputs exactly match the true data distribution, thereby it can also be used to measure the calibration performance. In multi-

class classification setting, NLL is equivalent to the cross-entropy loss, which is a commonly used loss function in deep learning. It should be noted that different from ECE, NLL takes the ground-truth label of each individual sample into account in evaluation, and hence would be influenced by the model's predictive performance. For example, a model cannot yield high NLL with poor accuracy even that it achieves zero ECE.

## 2.2 CALIBRATION OF DEEP NEURAL NETWORKS

**Post-hoc calibration.** Post-hoc approaches aim to recalibrate the model outputs after the training phase, for which scaling-based approaches are widely used due to their simplicity. By scaling the logit vector into a new vector wich same dimension, the sharpness of output probabilities can be changed. Formally, given a logit vector $z$ with $C$ dimensions, the output confidence with scaling fucntion $\sigma(\bullet)$ can be expressed as: $\hat{p} = \max_i \frac{\exp(\sigma(z_i))}{\sum_{c=1}^{C} \exp(\sigma(z_c))}$. The scaling function is a parametric model and can be instantiated with different operations that involve different number of parameters, which need to be learned on validation set after main training. Temperature scaling involves only a single parameter called temperature $T$: $\sigma_{TS}(z) = z/T$; Platt scaling learns two scalar parameters $a$ and $b$: $\sigma_{PS}(z) = az + b$; Vector scaling and matrix scaling learn a specific scalar for each class with linear transformation $Wz + b$, where $W$ is a full matrix and restricted to be a diagonal matrix in matrix scaling and vector scaling respectively. Among these approaches, temperature scaling is often the most effective one and does not change the original predictions.

**Regularization for Calibration.** Recently, there is a surge of studies on model calibration by leveraging implicit or explicit regularization techniques during training of DNNs, which makes better calibrated predictions by avoiding the overconfident outputs. Weight decay, which is now a standard technique in modern deep learning, was found to significantly impact model calibration performance in the pioneering study (Guo et al., 2017). Label Smoothing, which is widely used as a means to reduce overfitting of DNNs, has been demonstrated can calibrates DNNs by preventing the networks from becoming overconfident. Patra et al. (2023); Joo & Chung (2021) proposed to leverage entropy maximization-based regularization terms in loss to explicitly penalize the overconfident outputs. There are also some implicit regularization methods that are initially proposed to improve generalization, have also been found beneficial for calibration, such as mixup training, self-distillation and focal loss (Thulasidasan et al., 2019; Gupta et al., 2021; Mukhoti et al., 2020).

Although these methods do help model calibration after main training, they have been found to not work well in conjunction with post-calibration techniques. Our experiments in the Introduction section revealed that this phenomenon is ubiquitous among regularization methods. There are only a few studies have focused on how to improve the calibratability of neural networks. Wang et al. (2023) proposed to mitigate the mixup's harm on calibratability by avoiding the implicit label smoothing in mixup operation. Zhu et al. (2023) introduced a output masking strategy to reduce the harm of label softening in self-distillation on calibratability. These studies merely consider the negative impact of specific regularization methods on calibratability and have not delved deep into the neural networks to investigate the underlying reasons for the decline in calibratability.

# 3 WHAT MAKES NEURAL NETWORKS LESS CALIBRATABLE?

## 3.1 CALIBRATABILITY OF INTERMEDIATE FEATURES

Given the significance of the calibratability, we certainly expect to design new method to enhance it for deep neural networks. Before doing this, it is crucial to first understand the reasons behind those specific failure cases on model calibratability. The illustrative experiments in Figure 1 demonstrates a correlation between calibratability and training duration. Moving forward, we can explore how the the spatial structure of neural networks impact calibratability. To this end, we begin with an experiment that assesses the calibratability of the intermediate features across the hidden layers from bottom to top of neural networks. Specifically, for a well trained neural network, we freeze its bottom $k$ layers, and perform linear probing on the features extracted from these frozen layers. We carried out this experiment on different neural network architectures. Besides the standard trained models, we also use the models trained with large weight decay and labels smoothing, since these two regularization approaches often yield worse calibratability than standard training. For linear

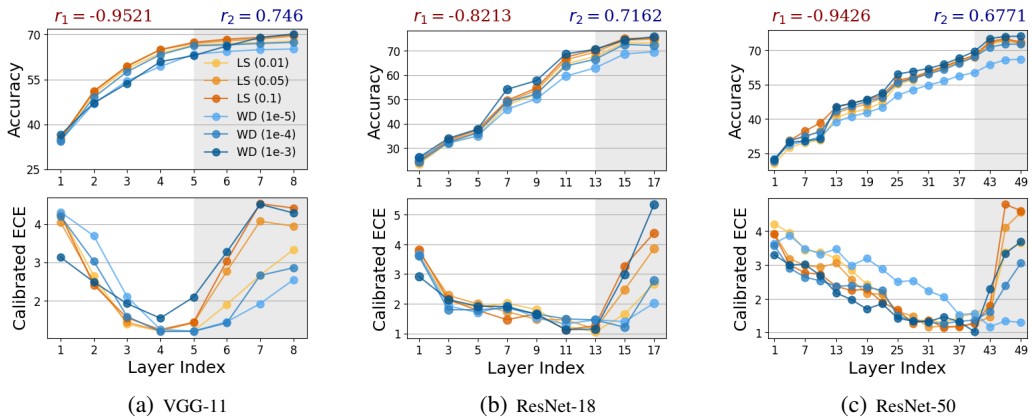

(a) VGG-11 (b) ResNet-18 (c) ResNet-50

Figure 2: The calibrated ECE and accuracy of linear probing over hidden layers on CIFAR-100 with different model architectures. The models are trained with weight decay and label smoothing using different regularization strengths. The red and blue texts at the top represent the Pearson correlation coefficients between accuracy and calibrated ECE of the bottom and top layers, respectively.

probing, we train the additional linear classifier on different intermediate layers with the same implementation. After that, we perform post-calibration and evaluate the calibrated ECE and accuracy of for these linear classifiers. For other experimental details such as the choice of optimizer, learning rate scheme and layer index for linear probing, please refer to Appendix A.1.

Figure 2 and Figure 8 show the results on CIFAR-100 and CIFAR-10. Each curve represents the result of linear probing on a specific hidden layers of the same neural network. We can observe that as the layer depth increases, the accuracy of linear probing gradually increases. However, the calibrated ECE follows an intriguing trend: The bottom layers generally yield quite large calibrated ECE, then as we move deeper into the networks it tends to decrease; Then, the deepest few layers, those near the top of neural networks, exhibit a striking raising on calibrated ECE, while these layers contribute little to the predictive performance. This calibration collapse phenomenon is observed across various network architectures, although the numbers of those specific layers at which the calibration performance starts to deteriorate may vary depending on architectures.

**Correlation between calibrated ECE and accuracy.** As mentioned in Introduction section, there is a trade-off between the discriminablility and calibratability especially when employing regularization in training. Here, we can observe that by dividing a neural network into two parts containing bottom layers and top layers, the correlation between these two properties can be decoupled. For the bottom (top) layers, accuracy and calibrated ECE exhibit a clear negative (positive) correlation. More importantly, we find that the most top layers offer minimal gains in terms of accuracy while inflicting significant damage to calibration. For instance, in Table 1, we present the results of linear probing for features from several top layers of ResNet-18 on CIFAR-100.

The accuracy and calibrated ECE that yiled significant changes compared to the previous layer are highlighted using colors. We can observe that for all the weight decay policies, the top layers significantly increase the calibrated ECE with limited accuracy gain, and the stronger the weight decay, the more pronounced this phenomenon becomes. It is worth noting that for training with a weight decay of $10^{-4}$, the features that achieve the best performance in both discriminability and calibratability are extracted from the 15th layer rather than the penultimate layer. Table 4 in Appendix shows this can be observed from different datasets and architectures.

Table 1: Accuracy and calibrated ECE of linear probing on the top layers of ResNet-18 on CIFAR-100.

| Layer Index | | 13 | 15 | | 17 | |
|---|---|---|---|---|---|---|
| WD ($10^{-3}$) | Acc | 71.5 | 74.7 | 3.2↑ | 75.9 | 1.2↑ |
| | ECE | 1.04 | 2.86 | 1.8↑ | 5.03 | 2.2↑ |
| WD ($10^{-4}$) | Acc | 66.5 | 72.6 | 6.1↑ | 72.1 | 0.5↓ |
| | ECE | 1.45 | 1.21 | 0.2↓ | 2.78 | 1.6↑ |
| WD ($10^{-5}$) | Acc | 63.8 | 70.0 | 6.2↑ | 70.3 | 0.3↑ |
| | ECE | 1.29 | 1.33 | 0.0↓ | 2.25 | 0.9↑ |

**Information compression in neural networks.** We think the collapse of calibration performance in top layers is related to the information compression of neural networks. Shwartz-Ziv & Tishby (2017) and Saxe et al. (2019) revisited the information bottleneck principle (Tishby et al., 2000) to explain the training dynamics of deep networks. They found that during training, deep neural networks first fit the data and then compress the information carried by the hidden layer features,

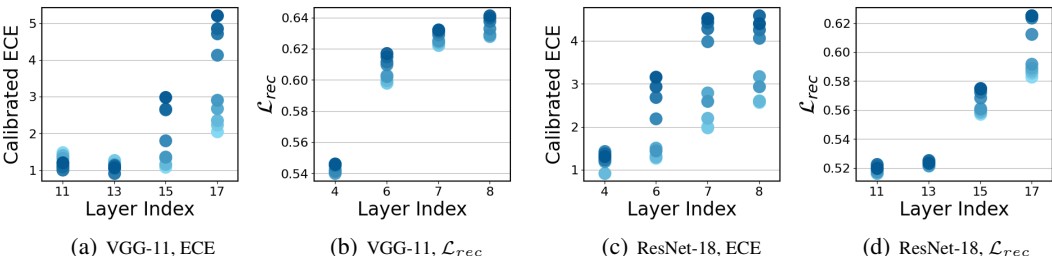

Figure 3: Calibrated ECE and reconstruction loss of models trained with different weight decay coefficients $(10^{-5}, 2.5\times10^{-5}, 5\times10^{-5}, 7.5\times10^{-5}, 1\times10^{-4}, 2.5\times10^{-4}, 5\times10^{-4}, 7.5\times10^{-4}, 1\times10^{-3}\})$ on CIFAR-100. Darker points represent models with larger weight decay.

with the compression intensity increasing with the layer depth. Specifically, the mutual information $I(x, z)$ between the original feature $x$ and the intermediate feature $h$ is use to measure the amount of retained information in $h$. Due to the intractability of the exact form of mutual information, which is not even feasible to approximately estimated by binning-based approximation in high dimensional feature space[1], we adopt the decoder-based estimation strategy by training a light-weight decoder and use the reconstruction loss to measure it (Wang et al., 2020). Similar to the linear probing experiments in the previous subsection, we build decoder upon the intermediate layers and fit it to the training data to measure the minimal reconstruction loss it can achieve. For more implementation details and a discussion on the rationale behind this strategy for estimating mutual information, please refer to the Appendix A.1.

Figure 3 presents the calibrated ECE of linear classifiers and the reconstruction loss of decoders on the top layers of VGG-11 and ResNet-18 trained with different regularization strengths. These two metrics demonstrate very similar trends over the top hidden layers. Taking ResNet-18 as an example, if we focus on the outputs from the 11th and 13th layer, all models achieve similar calibrated ECE and reconstruction loss. However, the models trained with larger weight decay yield more string raising for both calibration and reconstruction loss in the 15th and 17th layer. This demonstrate an unexpected negative impact of the information compression effect of neural networks for calibratability. As prior works on information compression of deep neural networks highlighted its benefits for improving predictive performance, this results further underscore concerns about the trade-off between calibratability and discriminability.

## 3.2 LEARNING CALIBRATABLE REPRESENTATIONS REQUIRES A WEAK CLASSIFIER

Having identified that the top layers of the neural network significantly compress the sample information, thereby reducing the model calibrability, we now explore how to prevent these layers from excessive information compression during training. As the compression and calibration degradation often occurs in the later stage of training, we firstly adopt a simple top-layer early stopping strategy to see if we can mitigate the harm of calibrability by avoiding overtraining of the top layers. Given a neural network, we select a few hidden layers as the cut points, and freeze the parameters of the layers upon these layers after the network is trained for a certain number of epochs. We conduct experiments on CIFAR-100 with VGG-11 and ResNet-18. The layer indices of the cut points and freezing epochs is presented in Figure 4. Each cell in the heatmaps displays the corresponding calibrated ECE and accuracy at the end of training with specific each top-layer early stopping policy.

As illustrated in the Figure 4, the cut points of the hidden layers and the number of freezing epochs have significant impacts on both the calibrated ECE and accuracy. Generally speaking, freezing the parameters of the top layers later improves accuracy but reduces calibrability. On the other hand, a larger freezing ratio enhances model calibrability but compromises its accuracy. It is shown that some specific top-layer early stopping policies can achieve a quite good trade-off between accuracy and calibrated ECE that surpasses the performance of full training. As is highlighted with a red and blue boxes, freezing the parameters of the 16th and above layers of ResNet-18 before the 30th epoch, and the 5th and above layers of VGG-11 at the 50th epoch can yield impressive results. If we consider the top layers as the classification module and the bottom layers as the representation

---

[1]It is found that binning-based calculation of mutual information is highly sensitive to the choice of activation functions used in the hidden layers, and some of the claims in (Shwartz-Ziv & Tishby, 2017) do not hold true in the general case (Saxe et al., 2019; Lorenzen et al., 2021).

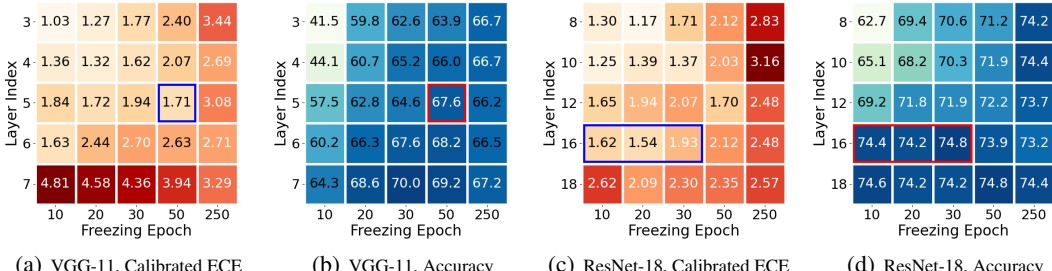

(a) VGG-11, Calibrated ECE  (b) VGG-11, Accuracy  (c) ResNet-18, Calibrated ECE  (d) ResNet-18, Accuracy

Figure 4: The heatmaps of calibrated ECE and accuracy on CIFAR-100 with a series of top-layer early stopping strategies. **Warmer colors** indicate poorer calibratability, while **cooler colors** signify better predictive performance.

module, this empirically justify a *weak classifier hypothesis*: Given a frozen classifier that has not been overtrained, the representation module can be learned better in terms of producing more calibratable features. While the performance of this top-layer early stopping strategy is very sensitive to the choices of layer index and freezing epochs, thus far, we have been ready to design a more advanced method based on this observation.

**Progressively Layer-peeled Training (PLT).** Now, we describe a straightforward method called PLT to better exploit the weak classifier hypothesis, which is expected to improve calibration without sacrificing accuracy. Our experiments suggest that finding an optimal results by freezing a specific portion of parameters at a given frozen epoch is challenging, given the myriad of possible combinations of layer cut points and epochs. To address this, PLT gradually frozen the hidden layers from top to down during the whole training procedure. Specifically, for a neural network with $L$ layers, we first divide it into $K$ ($K <= L$) parts, for which the cut points can lies between every adjacent layers or two adjacent parts containing several layers, such as two adjacent blocks in ResNets. Then, we partition the training duration into $K$ phases and gradually freeze the parameters of the exposed top layers as the training progresses. The number of training phases is same with that of the divided layer groups, thereby all groups can be exposed as the top trainable part for a certain epochs. For the number of epochs in each training phase $T_k$, $k \in \{1, 2, ..., K\}$, we can evenly partition the total number of epochs into $K$ phases, or set other partitioning strategies to adjust the timing of freezing the parameters of the top layers.

Here, we adopted a simple heuristic partitioning strategy: For the $k$th phase, we determine its training epochs as: $T_k = \frac{k^\gamma - (k-1)^\gamma}{K^\gamma} \cdot T$, where $\gamma$ is a hyperparameter used to control the distribution of training epochs across all phases. With $\gamma$ equal to 1, the total training epochs will evenly partitioned into all phases. With $\gamma$ greater than 1, start frozen epoch for all groups during training will occur earlier and vice versa. Figure 5 shows an example that divides the neural network into three parts. In this work, we set $\gamma$ to 1 for the main experiments and conduct empirical study for the choice of $\gamma$.

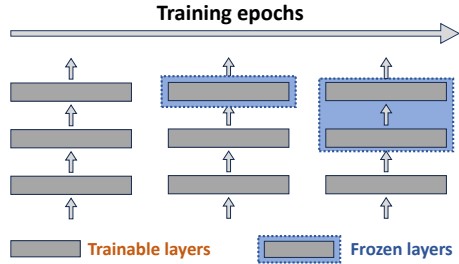

Figure 5: The PLT Method.

**Weight decay for frozen layers.** It is crucial to emphasize that for the parameters that are frozen, we continue to apply the same weight decay operation as for the trainable layers. We empirically found that maintaining this operation has minimal impact on calibration performance. However, it improves the predictive performance and makes our method perform on par with full training. Surprisingly, despite our method essentially being a straightforward combination of top-down training and early stopping, to the best of our knowledge, no existing research has highlighted the advantages of this training strategy for either accuracy or calibration. We speculate that the neglect in applying weight decay to the parameters of the frozen layers could be a reason why this strategy has not been fully exploited to its potential in previous works.

**Other related approaches.** There are several existing works employing the top-down training strategy. Fang et al. (2021) introduced the layer-peeled model, from which we actually drew inspiration for the name of our method, as a nonconvex yet analytically tractable optimization framework to better understand deep neural networks. Based on this model, Yang et al. (2022) proposed to learn a neural network from class-imbalanced datasets, with the classifier (i.e., the last fully connected layer

Table 2: The comparative results of several metrics (calibrated ECE, calibrated NLL and Accuracy) with ResNet-18. The number in each bracket indicates the ranking across all methods.

| | Metrics | Standard | WD | LS | SD | MT | FL | MMCE | Brier | MIT | **PLT** |
|---|---|---|---|---|---|---|---|---|---|---|---|
| SVHN | ECE | 0.44 (1) | 0.63 (4) | 0.96 (8) | 1.06 (9) | 1.95 (10) | 0.59 (3) | 0.93 (7) | 0.55 (2) | 0.83 (6) | 0.74 (5) |
| | NLL | 0.17 (5) | 0.15 (1) | 0.17 (6) | 0.19 (7) | 0.23 (9) | 0.16 (3) | 0.19 (8) | 0.17 (4) | 0.31 (10) | 0.15 (2) |
| | Accuracy | 95.5 (6) | 96.2 (1) | 95.7 (3) | 95.4 (7) | 94.5 (9) | 95.7 (4) | 95.1 (8) | 95.6 (5) | 91.7 (10) | 96.0 (2) |
| CIFAR-10 | ECE | 0.83 (2) | 1.28 (8) | 1.46 (9) | 1.54 (10) | 1.13 (7) | 1.13 (6) | 1.06 (5) | 1.03 (4) | 0.68 (1) | 0.85 (3) |
| | NLL | 0.18 (4) | 0.19 (6) | 0.22 (10) | 0.21 (9) | 0.17 (3) | 0.19 (7) | 0.20 (8) | 0.19 (5) | 0.16 (1) | 0.16 (2) |
| | Accuracy | 94.4 (7) | 94.6 (5) | 94.7 (3) | 94.4 (8) | 95.9 (1) | 93.9 (10) | 94.2 (9) | 94.4 (6) | 94.9 (2) | 94.7 (4) |
| CIFAR-100 | ECE | 2.61 (6) | 5.27 (9) | 4.41 (8) | 6.99 (10) | 1.18 (1) | 1.19 (2) | 2.28 (5) | 3.31 (7) | 2.03 (4) | 1.59 (3) |
| | NLL | 0.99 (6) | 0.96 (5) | 1.05 (9) | 1.07 (10) | 0.92 (4) | 0.92 (3) | 1.03 (8) | 1.00 (7) | 0.89 (1) | 0.91 (2) |
| | Accuracy | 73.8 (9) | 76.5 (2) | 76.2 (3) | 75.6 (4) | 76.9 (1) | 74.2 (8) | 72.5 (10) | 74.4 (7) | 75.6 (5) | 75.2 (6) |
| Tiny-ImageNet | ECE | 1.20 (2) | 1.77 (7) | 1.81 (8) | 1.24 (4) | 1.53 (6) | 3.16 (10) | 1.24 (3) | 2.59 (9) | 1.38 (5) | 1.18 (1) |
| | NLL | 2.26 (6) | 2.23 (4) | 2.39 (10) | 2.12 (2) | 2.26 (7) | 2.27 (8) | 2.29 (9) | 2.24 (5) | 2.22 (3) | 1.96 (1) |
| | Accuracy | 48.2 (7) | 48.1 (8) | 49.0 (6) | 50.4 (2) | 49.9 (3) | 47.1 (10) | 47.4 (9) | 49.2 (4) | 49.1 (5) | 52.0 (1) |
| Average Ranking | | 5.08 | 5.00 | 6.92 | 6.83 | 5.08 | 6.17 | 7.42 | 5.42 | 4.42 | **2.67** |

| Size | Pre-train | Metrics | Standard | WD | LS | MT | **PLT** |
|---|---|---|---|---|---|---|---|
| 64x64 | ✔ | ECE | 1.44 (2) | 1.88 (4) | 1.88 (5) | 1.66 (3) | 1.32 (1) |
| | | NLL | 1.99 (2) | 2.05 (4) | 2.11 (5) | 2.03 (3) | 1.83 (1) |
| | | Accuracy | 54.1 (4) | 53.2 (5) | 54.7 (2) | 54.4 (3) | 56.1 (1) |
| 224x224 | ✗ | ECE | 1.53 (2) | 2.31 (4) | 2.37 (5) | 1.85 (3) | 1.33 (1) |
| | | NLL | 1.69 (4) | 1.68 (3) | 1.80 (5) | 1.62 (2) | 1.50 (1) |
| | | Accuracy | 60.2 (4) | 59.9 (5) | 60.5 (3) | 62.7 (2) | 62.9 (1) |
| 224x224 | ✔ | ECE | 1.54 (2) | 3.08 (4) | 3.09 (5) | 2.65 (3) | 1.35 (1) |
| | | NLL | 1.37 (3) | 1.47 (5) | 1.46 (4) | 1.34 (2) | 1.27 (1) |
| | | Accuracy | 66.8 (4) | 65.4 (5) | 67.3 (3) | 68.4 (1) | 68.4 (2) |
| Average Ranking | | | 3.00 | 3.89 | 4.11 | 2.56 | **1.11** |

Table 3: The comparative results on Tiny-ImageNet.

Figure 6: Results with varied $\gamma$.

of a neural network) randomly initialized as an equiangular tight frame and fixed during training. Zhang et al. (2019) proposed a progressive top-down training method to alleviate the undertraining of the bottom layers. However, there is fundamental difference between their method and ours: Their method is based on the *good classifier hypothesis*, i.e., given a fixed classifier that is sufficiently well trained, the feature representation layers can be further enhanced to fit that good classifier. Moreover, their method employs the retraining strategy with reinitialized bottom layers.

## 3.3 COMPARATIVE EXPERIMENTS

In this subsection we conduct comparative experiments on four image classification datasets: SVHN Netzer et al. (2011), CIFAR-10, CIFAR-100 Krizhevsky (2009) and Tiny-ImageNet Deng et al. (2009). All the reported results are based on the average of 5 random trials. The implementation details including dataset splitting, training policies and the introduction of comparison methods can be found in appendix A.1. Our code will be made publicly available.

**Comparison with other methods.** We first compare the proposed PLT method with several methods that have been found to benefit model calibration. Table 2 shows the accuracy as well as the calibrated ECE and NLL based on temperature scaling. Tables 7 to 9 in Appendix show the results with ResNet-50 and other post-hoc calibration approaches. It can be observed that our method achieves the best average results on these metrics and is highly robust to different datasets and architectures. Where standard training already achieves satisfactory calibration, our method largely maintains the performance, while in cases where standard training falls short in calibration, our approach offers notable enhancements. Conversely, the performance of the regularization methods is extremely unstable. While methods like mixup and focal loss perform well on certain datasets, their overall performance does not match ours. Remarkably, the trainable calibration loss MMCE, which is specifically designed to enhance calibration in training, significantly harms model calibratability. If we focus on the NLL, which reflects both predictive performance and calibration, our method has a significant advantage over all other methods. Mixup inference in training (MIT), the only method that considered model calibratability during its design among all comparison methods, achieves the best overall performance apart from ours. Table 3 and Table 5 in Appendix shows the results with different experimental settings on Tiny-ImageNet, including training models both from scratch and pre-trained parameters, as well as models with different input sizes. We found that although the performance of models trained under various settings differs markedly, our method consistently demonstrate superior performance compared with other methods regardless of the settings.

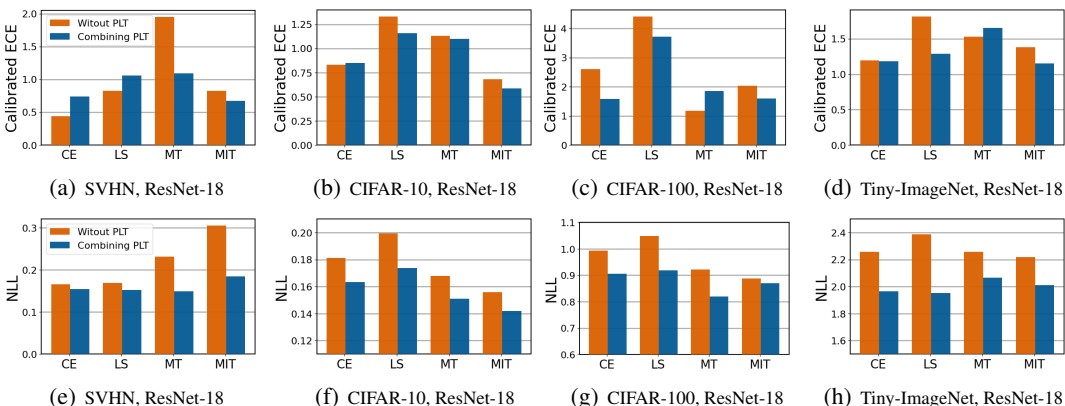

Figure 7: Top: The calibrated ECE by combining PLT and other methods. Bottom: The calibrated NLL by combining PLT and other methods.

**Hyperparameter sensitivity.** Our method includes a hyperparameter $\gamma$, which determines the starting epoch at which the top layers begin to be frozen. When $\gamma$ is smaller, top layers will start freezing later, and when it approaches 0, PLT becomes equivalent to standard training. As $\gamma$ increases, the number of epochs for training the top layers reduces. In the comparative experiments, we set $\gamma$ to 1 for all datasets. Here, we present the parameter sensitivity study. Overall, as shown in Figure 6, our method shows stable performance within a reasonable parameter range. Moreover, in cases like SVHN, where standard training already achieves relatively good calibratability, a smaller $\gamma$ yields better results. Conversely, for cases like CIFAR-100, where standard training obtains poor calibratability, a larger $\gamma$ (i.e., earlier parameter freezing) yields better results. This contrasting comparison further confirms the weak learner hypothesis mentioned in Subsection 3.2.

**Combining with other methods.** Our strategy is straightforward and easily integrated with other methods. To verify this, we conduct experiments by combining PLT with other methods. As can be observed in Figure 7 and Figure 10 in Appendix, our method can enhance these methods for both model calibration and predictive performance. For instance, as previously mentioned, MIT can also improve model calibratability to some extent. By integrating it with our method, we can leverage its data augmentation capabilities, further boosting the model's performance in terms of both calibration and predictivie performance. For methods like LS and Mixup, which may impair model calibratability, integrating PLT can help reducing the there harm on model calibration.

Our experiments verified the effectiveness of PLT. This training strategy, by reducing a significant number of parameter updates, will also accelerate the training process. However, our experiments are more of an analytical study, trying to validate the weak learner hypothesis for model calibration. It's important to emphasize that the PLT strategy still has considerable room for further improvement. As indicated in the aforementioned experiments, by adjusting the hyperparameters, we can further enhance the performance of PLT. Therefore, designing adaptive strategies to determine the starting frozen epoch for hidden layers holds significant promise for future research.

## 4 CONCLUSIONS

In this work, we shifted our focus from conventional calibration challenges during training to investigate model calibratability. To investigate the reasons for the decline in neural networks's calibratability, we delve into the calibratability of the intermediate features of the hidden layers of neural networks. Our experiments revealed that the calibratability challenge lies in the overtraining of the neural network's top layers, which suggests that for better calibratability, a more restrained classification head on the top of a neural network might be warranted. This observation is related to the information compression of deep neural networks, while sheds light on the unforeseen drawbacks of excessive information compression on uncertainty calibration. In overcome the calibratability issue, we introduced a progressively layer-peeled training strategy that focuses on mitigating the pitfalls associated with overtraining and information compression of top layers. By gradually freezing the parameters of the top layers during training, our method ensures these layers retain critical information, thus enhancing model calibratability.

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

# A APPENDIX

## A.1 EXPERIMENTAL DETAILS

**Datasets and training details.** For Tiny-ImageNet, the experiments are based on images resized as 64x64 and 224x224 repectively. We split the original training dataset into training set and validation set for main training and post-hoc calibration with the following ratios: 68257/5k for SVHN, 45k/5k for CIFAR-10/100 and 90k/10k for Tiny-ImageNet. We train VGG-11, ResNet-18 and ResNet-50 on these datasets. For Tiny-ImageNet, in addition to training from scratch, we also conduct training based on the open sourced pre-trained models. We use SGD as the opimizer with a momentum of 0.9 and a weight decay of $10^{-4}$ unless otherwise specified. We train on SVHN/CIFAR-10/CIFAR-100 by total 350 epochs with the initial learning rate as 0.1, and divide it by a factor of 10 after 150 epochs and 250 epochs respectively. For training from scratch on Tiny-ImageNet, we train 200 epochs with the initial learning rate as 0.1, and divide it by a factor of 2 after every 30 epochs. For training based on pretrained models on Tiny-ImageNet, we set the initial learning rate as 0.01 and use the same training policy with training from stratch. We set the batch size as 128 on SVHN/CIFAR-10/CIFAR-100, and 64 on Tiny-ImageNet. For post-hoc calibration, we adopt temperature scaling in main experiments and also present the results with platt scaling and vector scaling in Tables 8 and 9.

**Linear probing on intermediate features.** In Figure 2, we conduct linear probing on the features extracted over hidden layers. Here, we train the linear classifier by using SGD with a momentum of 0.9 and a weight decay of $10^{-4}$. We train the classifier for 10 epochs with the initial learning rate as 0.1, and decay it by a factor of 10 after 6 and 8 epochs respectively.

**Decoder-based mutual information estimation.** In Figure 3, we estimate the mutual information $I(x,h)$ by training a decoder parameterized by $w$ to obtain the minimal reconstruction loss on training data. As is discussed in (Wang et al., 2020), the mutual information can be bound by: $I(x,h) = H(x) - H(x \mid h) \geq H(x) - \mathcal{R}(x \mid h)$, where $\mathcal{R}(x|h)$ denotes the expected reconstruction error and $H(x)$ denotes the marginal entropy of $x$, as a constant. Therefore, we can estimate $\mathcal{R}(x|h)$ by training a decoder to measure the minimal reconstruction loss: $I(h,x) \approx \max_w [H(x) - \mathcal{R}_w(x \mid h)]$. In Figure 3, we directly use the binary cross-entropy loss on original feature space to reflect the mutual information. We train a light-weight decoder with two convolutional layers using Adam optimizer for 30 epochs with a constant learning rate 0.01. We are primarily focus on the comparisons of information across intermediate layers and between different models rather than obtaining the exact values of $I(x,h)$. Therefore, the utilization of the same training policy among all layers or models makes the comparisons fair.

**Layer cut points used for PLT.** As we discussed in Section 3.2, the PLT method needs to first divide all the layers into multiple parts, for which the cut points can lies between the adjacent groups containing several adjacent layers. Here, we divide ResNet-18 and ResNet-50 into 11 parts with the cut point sets $\{2, 4, 6, 8, 10, 12, 14, 16, 17, 18\}$ and $\{11, 17, 23, 26, 29, 32, 38, 41, 44, 47\}$, respectively.

**Details of comparison methods.** We compared several methods in the experiments of the main text: (large) weight decay (WD), label smoothing (LS), self-distillation (SD), mixup (MT), focal loss (FL), Maximum Mean Calibration Error (MMCE), Brier and mixup in training (MIT). Among these methods, the former five ones are not original designed for calibration but has been found beneficial to it in recent studies. MMCE is a specifically designed loss function term for calibration. MIT is proposed to mitigate the harm of the original mixup for calibration. Brier loss is the simple mean squared error between the predicted confidences and the ground-truth one-hot labels, which was considered as an important baseline as it can be decomposed into calibration and refinement (DeGroot & Fienberg, 1983). For all these methods, we adopt the same optimizer, learning rate and the number of epochs in training as discussed at the beginning of Appendix. Most of these methods involve hyperparameters that need to be determined before training:

- Label smoothing is widely used to reduce overfitting of DNNs (Szegedy et al., 2016). The mechanism of LS is simple: when training with CE loss, the one-hot label vector $y$ is replaced with *soft* label vector $\widetilde{y}$, whose elements can be formally denoted as $\widetilde{y_i} = (1 - \epsilon)y_i + \epsilon/K, \forall i \in \{1, ..., K\}$, where $\epsilon > 0$ is a strength coefficient.

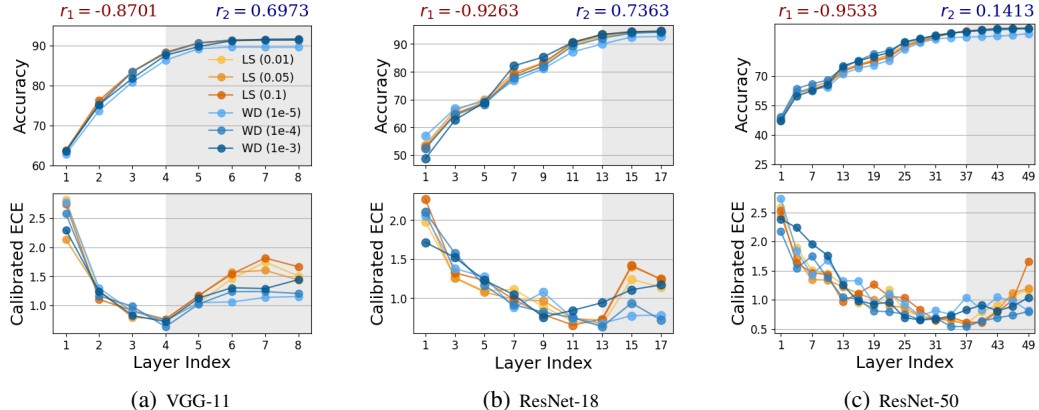

Figure 8: The calibrated ECE and accuracy of linear probing over hidden layers on CIFAR-10 with different model architectures. The models are trained with weight decay and label smoothing using different regularization strengths. The red and blue texts at the top represent the Pearson correlation coefficients between accuracy and ECE of the bottom and top layers, respectively.

- Mixup takes the convex combinations between pairs of examples and their labels: $\widetilde{x}_i = \lambda x_i + (1-\lambda)x_j, \widetilde{y}_i = \lambda y_i + (1-\lambda)y_j$, where $\lambda$ is sampled from $\mathrm{Beta}(\alpha,\alpha)$ with $\alpha > 0$. A larger $\alpha$ will result in a higher degree of mixing strength, thus making the mixed labels smoother.

- Focal loss is originally proposed to address the class imbalance problem in object detection.It is formally defined as: $\mathcal{L}_f = -(1-f_y^\theta)^\gamma \log f_y^\theta$, where $\gamma$ is a predefined coefficient. Mukhoti et al. (2020) found that the models learned by focal loss produce output probabilities which are already very well calibrated.

- MMCE is a continuous and differentiable proxy for calibration error and is normally used as a regularizer alongside the commonly used cross-entropy loss, where a weighting factor $\beta$ could be used to balance the contribution of MMCE (Kumar et al., 2018).

- Brier loss is the simple the mean squared error between the predicted confidences and the ground-truth one-hot labels, which was considered as an important baseline as it can be decomposed into calibration and refinement (DeGroot & Fienberg, 1983).

- MIT is a derived version of mixup for improving model calibration. The hyperparameter $\gamma$ plys the same role with that of the original mixup.

In Table 2, the hyperparameters for the comparison methods are chosen based on commonly recommended values in the literature: a decay ratio of $5 \times 10^{-4}$ for weight decay, $\epsilon$ as 0.1 for label smoothing, $\alpha$ as 1 for mixup and MIT, $\gamma$ as 3 for focal loss and $\beta$ as 0.5 for MMCE.

## A.2 COMPLEMENTARY EXPERIMENTS OF INTERMEDIATE FEATURES

In the main text, we present the empirical study on the calibratability of the intermediate features on CIFAR-100. Here, we present the results on CIFAR-10 in Figure 8. It should be noted that due to the differences in dataset difficulty, the calibration performance on CIFAR-10 is much better than that on CIFAR-100. However, as we can see, the detrimental impact of the top layers on calibratability observed in the main text still remains. Furthermore, the number of layers causing a decline in calibratability is consistent with the experiment on CIFAR-100, but the degree of degradation is somewhat less pronounced on CIFAR-10. Similarly, the utilization of regularization methods can lead to a deterioration in the calibratability of the intermediate features.

In Table 4, we present results reflecting the extent to which the top few layers enhance or hurt predictive performance and calibratability. The results of ResNet-18 are presented in Table 1 of the main text. The phenomenon that the most top layers offer minimal gains in terms of accuracy while inflicting significant damage to the calibration performance can be observed with different model architectures. In the reported cases, the representations that achieve the best performance in both discriminability and calibratability are not extracted from the penultimate layer. Furthermore,

Table 4: Accuracy (%) and calibrated ECE (%) of linear probing on the top layers of ResNet-50 (left) and VGG-11 (right) on CIFAR-100.

| Layer Index | | 4 | 6 | 7 | 8 |
|---|---|---|---|---|---|
| WD ($10^{-3}$) | Acc | 63.6 | 66.3 2.7↑ | 68.7 2.5↑ | 70.1 1.4↓ |
| | ECE | 1.91 | 3.02 1.1↑ | 4.41 1.4↑ | 4.28 0.1↓ |
| WD ($10^{-4}$) | Acc | 66.2 | 66.5 0.3↑ | 67.0 0.5↓ | 67.4 0.4↑ |
| | ECE | 1.20 | 1.42 0.2↓ | 2.66 1.2↑ | 2.87 0.2↑ |
| WD ($10^{-5}$) | Acc | 64.4 | 65.1 0.7↑ | 65.7 0.6↑ | 65.8 0.1↑ |
| | ECE | 1.20 | 1.35 0.2↓ | 2.10 0.8↑ | 2.59 0.5↑ |

| Layer Index | | 40 | 43 | 46 | 49 |
|---|---|---|---|---|---|
| WD ($10^{-3}$) | Acc | 7.03 | 75.6 5.3↑ | 76.7 1.1↑ | 76.6 0.1↓ |
| | ECE | 1.06 | 2.03 1.0↑ | 3.42 1.4↑ | 3.69 0.3↑ |
| WD ($10^{-4}$) | Acc | 67.1 | 71.3 4.2↑ | 72.6 1.3↑ | 72.6 0.0↓ |
| | ECE | 1.37 | 1.63 0.3↓ | 2.38 0.8↑ | 3.05 0.7↑ |
| WD ($10^{-5}$) | Acc | 61.9 | 65.6 3.7↑ | 67.3 1.7↑ | 67.6 0.3↑ |
| | ECE | 1.47 | 1.26 0.2↓ | 1.48 0.2↑ | 1.64 0.2↑ |

Table 5: The comparative results on Tiny-ImageNet with ResNet-50. We compare based on models both trained from scratch and pre-trained parameters, as well as models with different input sizes. The number in each bracket indicates the ranking across all methods.

| Size | Pre-train | Metrics | Standard | WD | LS | MT | **PLT** |
|---|---|---|---|---|---|---|---|
| 64x64 | ✔ | ECE | 1.65 (2) | 2.62 (4) | 3.81 (5) | 2.49 (3) | 0.94 (1) |
| | | NLL | 1.70 (2) | 1.76 (4) | 1.79 (5) | 1.70 (3) | 1.58 (1) |
| | | Accuracy | 60.2 (4) | 59.7 (5) | 62.3 (1) | 62.1 (2) | 61.3 (3) |
| 224x224 | ✗ | ECE | 1.73 (1) | 2.53 (4) | 3.03 (5) | 1.82 (2) | 1.87 (3) |
| | | NLL | 1.63 (2) | 1.68 (4) | 1.73 (5) | 1.53 (1) | 1.64 (3) |
| | | Accuracy | 61.9 (3) | 60.0 (4) | 61.9 (2) | 64.7 (1) | 59.4 (5) |
| 224x224 | ✔ | ECE | 2.33 (3) | 4.43 (5) | 3.88 (4) | 2.30 (2) | 1.89 (1) |
| | | NLL | 1.19 (3) | 1.32 (5) | 1.24 (4) | 1.16 (2) | 1.04 (1) |
| | | Accuracy | 71.8 (4) | 69.2 (5) | 72.6 (3) | 73.4 (2) | 74.0 (1) |
| Average Ranking | | | 2.67 | 4.44 | 3.78 | **2.00** | 2.11 |

when using a larger weight decay, the depth of the network layer that impairs model calibratability is shallower.

## A.3 COMPLEMENTARY RESULTS OF MAIN EXPERIMENTS

We present some complementary results of the experiments in Subsection 3.3.

- As we discussed in Section 3.2, for the parameters that are frozen, we continue to apply the same weight decay operation as for the trainable layers. We empirically found that maintaining this operation has minimal impact on calibration performance. The results are shown in Figure 9, as we can observe, this operation leads to favorable predictive performance with limited ECE increasing. As our goal is to achieve good calibration without the loss on predictive performance, we maintain this operation in our main experiments.

- The comparison of raw ECE between different methods is shown in Table 6. The experimental results indicate a significant advantage in the average performance of our method in terms of raw ECE. As you mentioned, although train-time calibration is not our primary objective, this excellent performance makes our method more practical in specific scenarios. It is noteworthy that despite the fact that the comparative methods are all train-time calibration strategies, their performance is subpar, which aligns with the results in our Figure 1 (where their performance exhibits a V-shaped trend in terms of hyperparameters). This is due to their requirement for highly precise hyperparameter strategies to achieve good

Table 6: The comparative results of **raw ECE** with ResNet-18 (top) and ResNet-50 (bottom).The number in each bracket indicates the ranking across all methods.

| | Standard | WD | LS | SD | MT | FL | MMCE | Brier | MIT | **PLT** |
|---|---|---|---|---|---|---|---|---|---|---|
| SVHN | 2.10 (5) | 1.34 (3) | 8.41 (7) | 33.14 (10) | 15.09 (9) | 12.31 (8) | 2.59 (6) | 0.59 (1) | 2.07 (4) | 1.06 (2) |
| CIFAR-10 | 3.34 (6) | 2.90 (4) | 7.29 (7) | 28.38 (10) | 10.16 (9) | 8.60 (8) | 3.02 (5) | 2.15 (3) | 1.17 (1) | 1.76 (2) |
| CIFAR-100 | 11.23 (6) | 4.80 (3) | 13.40 (8) | 41.28 (10) | 13.96 (9) | 8.76 (5) | 12.30 (7) | 4.55 (2) | 7.73 (4) | 3.62 (1) |
| Tiny-ImageNet | 21.62 (10) | 18.21 (8) | 1.71 (1) | 14.00 (7) | 4.84 (4) | 2.99 (3) | 20.20 (9) | 12.87 (5) | 13.25 (6) | 2.77 (2) |
| Average Ranking | 6.75 | 4.50 | 5.75 | 9.25 | 7.75 | 6.00 | 6.75 | 2.75 | 3.75 | **1.75** |
| SVHN | 2.64 (5) | 2.43 (4) | 7.60 (7) | 31.89 (10) | 14.83 (9) | 9.92 (8) | 2.72 (6) | 1.30 (1) | 1.58 (3) | 1.36 (2) |
| CIFAR-10 | 3.89 (7) | 3.35 (5) | 6.98 (8) | 22.36 (10) | 11.16 (9) | 3.70 (6) | 3.28 (4) | 2.85 (3) | 1.26 (1) | 2.60 (2) |
| CIFAR-100 | 12.49 (9) | 8.53 (7) | 4.94 (1) | 36.73 (10) | 7.30 (6) | 4.98 (2) | 12.41 (8) | 6.17 (4) | 7.04 (5) | 6.06 (3) |
| Tiny-ImageNet | 16.59 (9) | 18.55 (10) | 6.29 (3) | 9.76 (6) | 3.68 (2) | 2.91 (1) | 15.60 (8) | 13.28 (7) | 9.09 (5) | 7.69 (4) |
| Average Ranking | 7.50 | 6.50 | 4.75 | 9.00 | 6.5 | 4.25 | 6.50 | 3.75 | 3.50 | **2.75** |

(a) SVHN    (b) CIFAR-10    (c) CIFAR-100    (d) Tiny-ImageNet

(e) SVHN    (f) CIFAR-10    (g) CIFAR-100    (h) Tiny-ImageNet

Figure 9: Accuracy with and without WD on frozen layers.

train-time calibration performance. By the way, we think these results further validate the inadvisability of pursuing train-time calibration.

- Table 5 shows the results with different settings on Tiny-ImageNet, including training models from scratch and fine-tuning on pre-trained bases, as well as experiments with different input sizes. As is shown, our PLT method demonstrates superior performance. Initializing with pre-trained models and using larger image sizes greatly enhance model performance for all methods compared with the results in Table 2 in main text. However, regardless of the settings, our method consistently demonstrated superior performance than the others.

- Table 7 shows the results with ResNet-50 under the same setup with Table 2. Tables 8 and 9 show the results with both ResNet-18 and ResNet-50 by using platt scaling and vector scaling in post-hoc calibration stage. Although those two approaches involves more parameters needed to be tuned on validation data, they do not show significant improvement over temperature scaling on calibration. Regardless different post-hoc calibration approaches, the PLT method consistently outperforms other training methods in terms of overall performance.

- Figure 10 shows the results of calibrated ECE and accuracy by integrating PLT with several other methods. In most cases, our method can further enhance the model's calibratability and predictive performance based on other methods.

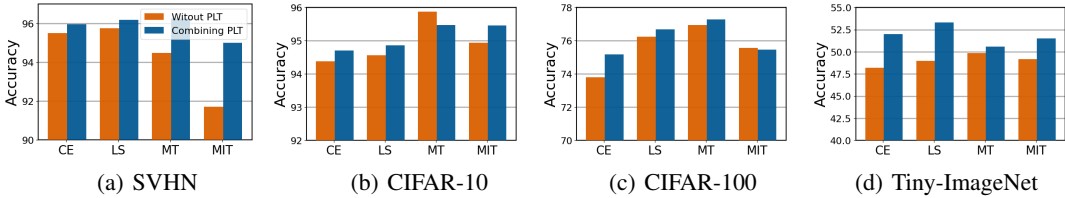

(a) SVHN  (b) CIFAR-10  (c) CIFAR-100  (d) Tiny-ImageNet

Figure 10: The calibrated ECE and accuracy of ResNet-18 by combining PLT and other methods.

Table 7: The comparative results of several metrics with ResNet-50. The number in each bracket indicates the ranking across all methods.

| | Metrics | Standard | WD | LS | SD | MT | FL | MMCE | Brier | MIT | **PLT** |
|---|---|---|---|---|---|---|---|---|---|---|---|
| SVHN | ECE | 0.64 (1) | 1.12 (4) | 1.71 (9) | 1.58 (8) | 2.11 (10) | 1.22 (6) | 1.17 (5) | 1.32 (7) | 0.93 (2) | 1.10 (3) |
| | NLL | 0.16 (1) | 0.17 (4) | 0.20 (9) | 0.20 (8) | 0.20 (7) | 0.17 (2) | 0.19 (6) | 0.18 (5) | 0.26 (10) | 0.17 (3) |
| | Accuracy | 95.7 (5) | 95.9 (1) | 95.8 (4) | 95.7 (6) | 95.5 (9) | 95.7 (7) | 95.6 (8) | 95.8 (3) | 93.2 (10) | 95.8 (2) |
| CIFAR-10 | ECE | 0.76 (4) | 1.15 (6) | 1.54 (9) | 1.51 (8) | 1.48 (7) | 1.12 (5) | 1.56 (10) | 0.70 (3) | 0.66 (1) | 0.67 (2) |
| | NLL | 0.18 (4) | 0.19 (6) | 0.25 (10) | 0.22 (9) | 0.16 (2) | 0.20 (8) | 0.20 (7) | 0.18 (5) | 0.15 (1) | 0.18 (3) |
| | Accuracy | 94.4 (7) | 94.5 (5) | 94.5 (4) | 94.3 (8) | 96.1 (1) | 93.5 (10) | 94.2 (9) | 94.5 (3) | 95.2 (2) | 94.4 (6) |
| CIFAR-100 | ECE | 2.73 (5) | 4.21 (8) | 5.08 (9) | 5.82 (10) | 2.77 (6) | 1.13 (2) | 2.10 (3) | 3.03 (7) | 2.21 (4) | 1.10 (1) |
| | NLL | 0.96 (6) | 0.93 (4) | 1.15 (9) | 1.18 (10) | 0.93 (3) | 0.92 (2) | 0.97 (7) | 1.06 (8) | 0.85 (1) | 0.94 (5) |
| | Accuracy | 74.6 (5) | 76.9 (2) | 75.1 (4) | 72.9 (9) | 77.5 (1) | 73.9 (6) | 73.5 (8) | 72.4 (10) | 76.7 (3) | 73.8 (7) |
| Tiny-ImageNet | ECE | 1.37 (5) | 1.79 (6) | 2.70 (9) | 1.34 (4) | 1.94 (7) | 3.28 (10) | 1.22 (3) | 2.54 (8) | 1.17 (1) | 1.19 (2) |
| | NLL | 2.05 (3) | 2.13 (8) | 2.22 (9) | 1.91 (2) | 2.05 (4) | 2.08 (5) | 2.10 (7) | 2.30 (10) | 2.09 (6) | 1.86 (1) |
| | Accuracy | 52.8 (5) | 50.5 (9) | 53.3 (4) | 55.2 (1) | 54.0 (3) | 50.9 (8) | 51.5 (6) | 47.9 (10) | 51.1 (7) | 54.5 (2) |
| Average Ranking | | 4.25 | 5.25 | 7.42 | 6.92 | 5.00 | 5.92 | 6.58 | 6.58 | 4.00 | **3.08** |

Table 8: The comparative results of ResNet-18 (top) and ResNet-50 (bottom) with **Platt Scaling**. The number in each bracket indicates the ranking across all methods.

| | Metrics | Standard | WD | LS | SD | MT | FL | MMCE | Brier | MIT | **PLT** |
|---|---|---|---|---|---|---|---|---|---|---|---|
| SVHN | ECE | 0.40 (1) | 0.61 (3) | 0.92 (8) | 1.07 (9) | 1.79 (10) | 0.62 (4) | 0.90 (7) | 0.51 (2) | 0.78 (6) | 0.73 (5) |
| | NLL | 0.17 (5) | 0.15 (1) | 0.17 (6) | 0.19 (7) | 0.23 (9) | 0.16 (3) | 0.19 (8) | 0.17 (4) | 0.30 (10) | 0.15 (2) |
| CIFAR-10 | ECE | 0.85 (2) | 1.25 (8) | 1.47 (9) | 1.50 (10) | 1.10 (6) | 1.15 (7) | 1.03 (4) | 1.09 (5) | 0.56 (1) | 0.86 (3) |
| | NLL | 0.18 (4) | 0.19 (5) | 0.22 (10) | 0.21 (9) | 0.17 (3) | 0.19 (7) | 0.19 (8) | 0.19 (6) | 0.15 (1) | 0.16 (2) |
| CIFAR-100 | ECE | 2.72 (6) | 5.26 (9) | 4.48 (8) | 6.79 (10) | 1.15 (1) | 1.33 (2) | 2.33 (5) | 3.43 (7) | 2.21 (4) | 1.63 (3) |
| | NLL | 1.00 (6) | 0.97 (5) | 1.05 (9) | 1.07 (10) | 0.91 (2) | 0.92 (4) | 1.04 (8) | 1.01 (7) | 0.88 (1) | 0.91 (3) |
| Tiny-ImageNet | ECE | 1.09 (1) | 1.56 (5) | 1.78 (8) | 1.19 (3) | 1.73 (7) | 2.54 (9) | 1.18 (2) | 2.79 (10) | 1.58 (6) | 1.38 (4) |
| | NLL | 2.24 (7) | 2.18 (4) | 2.37 (10) | 2.10 (2) | 2.21 (5) | 2.24 (8) | 2.26 (9) | 2.22 (6) | 2.18 (3) | 1.96 (1) |
| Average Ranking | | 4.00 | 5.00 | 8.50 | 7.50 | 5.38 | 5.50 | 6.38 | 5.86 | 4.00 | **2.88** |
| SVHN | ECE | 0.65 (1) | 1.05 (3) | 1.68 (9) | 1.62 (8) | 1.77 (10) | 1.20 (6) | 1.07 (4) | 1.30 (7) | 0.82 (2) | 1.16 (5) |
| | NLL | 0.16 (1) | 0.17 (4) | 0.20 (9) | 0.20 (8) | 0.19 (6) | 0.17 (3) | 0.19 (7) | 0.18 (5) | 0.26 (10) | 0.17 (2) |
| CIFAR-10 | ECE | 0.86 (4) | 1.19 (6) | 1.60 (10) | 1.58 (9) | 1.39 (8) | 1.14 (5) | 1.36 (7) | 0.74 (3) | 0.55 (1) | 0.70 (2) |
| | NLL | 0.18 (4) | 0.19 (6) | 0.25 (10) | 0.22 (9) | 0.16 (2) | 0.20 (8) | 0.19 (7) | 0.18 (5) | 0.15 (1) | 0.18 (3) |
| CIFAR-100 | ECE | 2.86 (6) | 4.34 (8) | 5.01 (9) | 5.91 (10) | 2.56 (5) | 1.50 (2) | 2.36 (4) | 3.20 (7) | 2.04 (3) | 1.20 (1) |
| | NLL | 0.96 (6) | 0.93 (4) | 1.15 (9) | 1.18 (10) | 0.91 (2) | 0.92 (3) | 0.98 (7) | 1.07 (8) | 0.84 (1) | 0.94 (5) |
| Tiny-ImageNet | ECE | 1.08 (1) | 1.63 (6) | 2.64 (9) | 1.41 (5) | 2.08 (7) | 3.25 (10) | 1.26 (3) | 2.60 (8) | 1.16 (2) | 1.28 (4) |
| | NLL | 2.03 (4) | 2.08 (8) | 2.20 (9) | 1.89 (2) | 2.01 (3) | 2.05 (6) | 2.08 (7) | 2.27 (10) | 2.04 (5) | 1.82 (1) |
| Average Ranking | | 3.38 | 5.63 | 9.25 | 7.63 | 5.38 | 5.38 | 5.75 | 6.63 | 3.13 | **2.88** |

Table 9: The comparative results of ResNet-18 with **Vector Scaling**. The number in each bracket indicates the ranking across all methods.

| | Metrics | Standard | WD | LS | SD | MT | FL | MMCE | Brier | MIT | **PLT** |
|---|---|---|---|---|---|---|---|---|---|---|---|
| SVHN | ECE | 0.38 (1) | 0.61 (3) | 0.92 (8) | 1.02 (9) | 1.76 (10) | 0.62 (4) | 0.88 (7) | 0.51 (2) | 0.77 (5) | 0.78 (6) |
| | NLL | 0.17 (5) | 0.15 (1) | 0.17 (6) | 0.19 (7) | 0.23 (9) | 0.16 (3) | 0.19 (8) | 0.17 (4) | 0.30 (10) | 0.15 (2) |
| CIFAR-10 | ECE | 0.89 (3) | 1.23 (8) | 1.52 (9) | 1.52 (10) | 0.94 (4) | 1.13 (7) | 1.01 (5) | 1.06 (6) | 0.56 (1) | 0.86 (2) |
| | NLL | 0.18 (4) | 0.18 (5) | 0.21 (10) | 0.21 (9) | 0.16 (3) | 0.19 (8) | 0.19 (7) | 0.18 (6) | 0.15 (1) | 0.16 (2) |
| CIFAR-100 | ECE | 2.96 (6) | 5.42 (9) | 4.58 (8) | 6.86 (10) | 1.54 (1) | 1.68 (2) | 2.69 (5) | 3.46 (7) | 2.10 (4) | 1.94 (3) |
| | NLL | 1.00 (6) | 0.97 (5) | 1.05 (9) | 1.07 (10) | 0.91 (2) | 0.93 (4) | 1.04 (8) | 1.01 (7) | 0.88 (1) | 0.92 (3) |
| Tiny-ImageNet | ECE | 1.67 (5) | 2.29 (8) | 2.69 (9) | 1.33 (2) | 2.24 (7) | 1.63 (4) | 1.43 (3) | 3.41 (10) | 1.98 (6) | 1.25 (1) |
| | NLL | 2.23 (7) | 2.18 (4) | 2.34 (10) | 2.11 (2) | 2.19 (5) | 2.23 (8) | 2.26 (9) | 2.21 (6) | 2.17 (3) | 1.96 (1) |
| Average Ranking | | 4.63 | 5.38 | 8.63 | 7.38 | 5.13 | 5.00 | 6.50 | 6.00 | 3.88 | **2.50** |
| SVHN | ECE | 0.89 (3) | 1.23 (8) | 1.52 (9) | 1.52 (10) | 0.94 (4) | 1.13 (7) | 1.01 (5) | 1.06 (6) | 0.56 (1) | 0.86 (2) |
| | NLL | 0.18 (4) | 0.18 (5) | 0.21 (10) | 0.21 (9) | 0.16 (3) | 0.19 (8) | 0.19 (7) | 0.18 (6) | 0.15 (1) | 0.16 (2) |
| CIFAR-10 | ECE | 0.82 (4) | 1.20 (6) | 1.21 (7) | 1.63 (10) | 1.31 (9) | 1.19 (5) | 1.26 (8) | 0.68 (2) | 0.54 (1) | 0.73 (3) |
| | NLL | 0.18 (5) | 0.19 (6) | 0.24 (10) | 0.22 (9) | 0.16 (2) | 0.20 (8) | 0.19 (7) | 0.18 (4) | 0.15 (1) | 0.18 (3) |
| CIFAR-100 | ECE | 3.12 (6) | 4.56 (8) | 5.08 (9) | 6.18 (10) | 2.64 (5) | 1.65 (2) | 2.42 (4) | 3.45 (7) | 2.24 (3) | 1.53 (1) |
| | NLL | 0.96 (6) | 0.93 (4) | 1.15 (9) | 1.18 (10) | 0.91 (2) | 0.93 (3) | 0.98 (7) | 1.07 (8) | 0.84 (1) | 0.95 (5) |
| Tiny-ImageNet | ECE | 1.43 (3) | 2.30 (7) | 3.06 (10) | 1.73 (4) | 2.81 (8) | 1.78 (6) | 1.20 (1) | 2.96 (9) | 1.77 (5) | 1.27 (2) |
| | NLL | 2.03 (4) | 2.08 (7) | 2.18 (9) | 1.89 (2) | 2.00 (3) | 2.04 (5) | 2.08 (8) | 2.26 (10) | 2.04 (6) | 1.83 (1) |
| Average Ranking | | 3.75 | 5.75 | 9.00 | 7.63 | 5.63 | 4.75 | 5.63 | 6.50 | 3.63 | **2.75** |

