# OpenReview forum: "Calibration Bottleneck: What Makes Neural Networks less Calibratable?"
_ICLR.cc/2024/Conference — Submitted to ICLR 2024_

### Official Review · Reviewer_VAxV · 2023-10-21

**Soundness:** 2 fair
**Presentation:** 2 fair
**Contribution:** 2 fair
**Rating:** 3
**Confidence:** 5

**Summary:**

This paper proposes the concept of calibratability, which refers to whether a trained model can achieve better calibrated performance after post-calibration. The authors conduct experiments to demonstrate the correlation between calibratability and model parameters. Based on this, they propose a progressive Layer-peeled Training strategy. Experiments are conducted to verify the effectiveness of the proposed method.

**Strengths:**

1. The experiments and analysis in the paper are thorough. For example, many experiments are conducted to demonstrate that previous regularization-based methods have poorer calibratability.
2. The calibratability problem studied in this paper is interesting.

**Weaknesses:**

1. The writing of the paper lacks clarity. For example, in academic writing, abbreviations should be explained when first introduced. However, the paper does not do this, leading to confusion about concepts and wasted time for me as a reader. For instance, abbreviations like WD, LS, MT appear in Figure 1 without explanation. The y-axis in Figures G and H should be calibrated ECE instead of plain ECE, right?
2. Despite extensive experiments analyzing why previous regularization-based methods have poorer calibratability, the paper fails to draw definitive conclusions. The various analyses only show correlation, not causation, between network depth and post-calibration ECE. Specifically, we can see depth is correlated with post-calibration ECE, but cannot conclude depth causes poorer post-calibration ECE. This is an issue with the motivation for the proposed method.
3. While the experimental analysis is thorough, the paper lacks theoretical analysis and guidance to lead to clear conclusions about factors influencing model calibratability.
4. The paper lacks formal definitions for key concepts like calibratability.
5. The performance of the proposed method is poor. In most cases, it does not even outperform plain training. When combined with plain training, it can even hurt ECE (the authors relegate this result to the appendix rather than the main text).

**Questions:**

The questions proposed in the Weaknesses.

---

> ### Author Response · Authors · 2023-11-16
> **Response to Reviewer VAxV**
>
> ***
> Thanks for your questions and insightful comments. The following are our responses:
> ***
>
> ***Q1.*** *`The writing of the paper lacks clarity. For example, in academic writing, abbreviations should be explained when first introduced.`*
>
> **A1.** Thanks for your feedback. We have made revisions to our paper based on the feedback from the reviewers, addressing aspects such as writing and organization.
>
> ***
>
> ***Q2.*** *`Despite extensive experiments analyzing why previous regularization-based methods have poorer calibratability, the paper fails to draw definitive conclusions. The various analyses only show correlation, not causation, between network depth and post-calibration ECE. Specifically, we can see depth is correlated with post-calibration ECE, but cannot conclude depth causes poorer post-calibration ECE. This is an issue with the motivation for the proposed method.`*
>
> **A2.** Thanks for the comment. However, we would like to sincerely defense the significance of conducting empirical studies on the calibration and calibratability problems. While there exists a substantial body of literature on train-time model calibration, the intricate interplay of these methods with post-hoc calibration has not been thoroughly investigated. We firmly believe that the primary empirical results about the correlation presented in this paper still contributes to the advancement of model calibration, since it provides strong evidence to support our claim that something may be amiss in the top few layers to achieve good calibratability. Please refer to additional discussions regarding the motivation in A1 of our responses to Reviewer x8eC.
>
> ***
>
> ***Q3.*** *`While the experimental analysis is thorough, the paper lacks theoretical analysis and guidance to lead to clear conclusions about factors influencing model calibratability. The paper lacks formal definitions for key concepts like calibratability.`*
>
> **A3.** Thanks for the comment. We acknowledge the inherent difficulty in theoretically studying the calibratability property of deep neural networks, an aspect our paper has not addressed, while also believe that investigating the problem through the lens of deep learning phenomena is an essential preliminary step to offer insights for future research in theoretic aspect. In this paper, we measure the calibratability by assessing the calibration performance after post-hoc calibration, i.e., given a specific post-hoc calibration approaches, the final calibration result a model can attain after recalibration. In this context, one can assert that models trained with, for example, smaller weight decay exhibit superior calibratability in contrast to those trained with stronger weight decay.
>
> ***
>
> ***Q4.*** *`The performance of the proposed method is poor. In most cases, it does not even outperform plain training. When combined with plain training, it can even hurt ECE (the authors relegate this result to the appendix rather than the main text).`*
>
> **A4.** Thanks for the comment. In some cases, PLT does not outperform standard training in terms of calibrated ECE. As discussed in the experimental section, for datasets like SVHN and CIFAR-10, standard training already demonstrates remarkable calibratability. In these scenarios, our method still remarkably achieves the trade-off between calibration and accuracy without the need to select specific parameters, compared with standard training and other train-time calibration methods. Additionally, it should be noted that the PLT strategy still holds substantial potential for improvement, for example in Figure 6, fine-tuning hyperparameters we can further enhance its performance. We also would like to contend that our experiments primarily serve as an analytical study aimed at identifying the causes leading to poor calibratability.

---

> > ### Comment · Reviewer_VAxV · 2023-11-23
> > **Official Comment by Authors**
> >
> > Thanks for your response. I will maintain my score for the following reasons:
> >
> > The author studies the concept of calibratability, but from beginning to end, this concept is neither mathematically nor formally defined.
> >
> > The author claims to have discovered several factors related to calibratability through extensive experiments, but these are merely correlations, not causations. I hope the author can understand. I can even list many factors related to calibratability without needing experiments.
> >
> > In the methods section, the author proposed some approaches to address the issue of calibratability, but the performance is not good in practice.
> >
> > The methodology and perspectives of the paper lack theoretical support and are merely subjective conjectures and heuristic designs.
> >
> > Based on the above considerations, I finally believe that this paper is not suitable for acceptance at this time. The authors should continue to improve the manuscript.

---

> > > ### Author Response · Authors · 2023-11-23
> > >
> > > Thanks for your feedback!
> > >
> > > We respect your opinions and the final score, although we may not agree with some of the points you raised regarding the experiments and correlation.
> > >
> > > Thank you again for your constructive feedback, and we will continue to improve this work.

---

> ### Author Response · Authors · 2023-11-23
>
> Dear Reviewer,
>
> We sincerely thank you again for your great efforts in reviewing this paper. We have gone through your points one-by-one and tried to address them carefully. Please don’t hesitate to let us know if you have any further questions and comments.
>
> Best regards,
>
> Authors

---

### Official Review · Reviewer_7Hnm · 2023-10-30

**Soundness:** 3 good
**Presentation:** 4 excellent
**Contribution:** 3 good
**Rating:** 8
**Confidence:** 4

**Summary:**

This paper investigates the issue of 'calibratability': specifically, which training techniques yield better calibration performance when combined with post-hoc calibration methods, and which techniques might lead to a decline in final calibration performance after recalibration. Building upon prior research, the authors found that common normalization techniques aimed at enhancing accuracy while penalizing overconfidence, i.e., weight decay, mixup, distillation, label smoothing, indeed hurts the final calibration (measured by ECE) when combined with temperature scaling (a post-hoc calibration method).

To study the problem, the authors analyzed the calibration performance of features from each layer using linear probing. They observed that for the initial layers, the calibration improves as training progressed. In contrast, the latter layers exhibited an increasing calibration error. The authors used the information bottleneck principle to interpret this phenomenon, suggesting that the initial layers of the model is fitting the training distribution, while the subsequent layers progressively compress model information to enhance the separability between classes. This compression process might compromise the model's calibratability (losing uncertainty information?).

Based on these observations, the authors proposed the 'weak classifier' hypothesis, which advocates for not overtraining the model's compression capability and not losing excessive information to preserve its favorable calibration performance, while still retaining methods like weight decay to maintain its accuracy. The implementation of this weak classifier involves gradually freezing the latter layers to ensure that the initial layers receive the most training, and the final layers receive the least.

**Strengths:**

**Originality:** 1) This paper introduces the concept of "calibratability". While prior works have touched upon some of its findings, this study offers more comprehensive empirical findings and insights. 2) While I am not deeply familiar with related works on "layer-peeled training," this appears to be the first paper emphasizing its role in calibration.

**Quality/Clarity:** This is a quite comprehensive and solid study that :
1. Introduces a pressing research question: Which kind of normalization techniques possess good calibratability?
2. Investigates this question through experiments to yield insightful empirical findings.
3. Explains these findings using information bottleneck principle, whose rationale fitting naturally and with empirical findings to support.
4. Proposes a solution based on these findings and understandings.
5. Verify this method through experiments.

**Significance:** The training method presented can enhance both calibration and accuracy, loosening the need for trade-offs and making it also practical.

**Weaknesses:**

**Quality/Significance:** The empirical findings and experiment results are based on resnet 18/resnet 50. Currently, more prevalent models lean towards vision transformers. From my observations, many transformer-based models behave differently from traditional resnet-type models in terms of calibration. For example, as highlighted in the "Revisiting the Calibration of Modern Neural Networks" study, models with better capacity like transformers tend to be more well-calibrated, while traditional models such as ResNet tend to be overconfident.

**Questions:**

Problem Definition & Empirical findings:
1. Defining calibratability as "how amenable a model is to be recalibrated in the post-training phase" seems a bit inappropriate? For instance, in figure 1a, after training using the student distillation method, the ECE error is significant. However, after applying temperature scaling, there's a considerable error reduction. From this perspective, it seems the model is calibratable. But because its performance after adding TS remains inferior to standard training + TS, the model appears less calibratable. I think the description of calibratability should be related to the final calibration performance after combining with a post-hoc calibration method?
2. It's interesting that figure 1 g/h have different tendencies. Do you have any intuition  why different dataset have different tendency for ECE dynamics?
3. Regarding that this compression process might compromise the model's calibratability", is it because the model, during the process of pushing each sample towards the class center, loses the uncertainty information of each sample in terms of their confidence?
4. Regarding post-hoc calibration methods, it seems you've only compared scaling-based methods. What about binning-based methods or kernel-density-estimation-based post-hoc calibration methods? Will they display similar behavior, and if this phenomenon also holds true with other methods? It's fine if there isn't time to do the experiments, I just feel it would be more solid to involve them.
5. I wonder if the reconstruction error is related to the dimension of the feature embedding of each layer. That is, if the feature embedding dimension begins to decrease e.g. from 4096 to 2048, will the model begin the compress their information? Or even the layer dimension remains 4096, it is still doing compression? I wonder whether we can infer from the dimension that at which layer the inflection point might occur? The thought behind this question is that to gain the best accuracy, whether the model will try to keep all the related information, even those unnecessary?

**Experimental section:**
1. Can the average ranking be calculated separately based on different metrics? It would make it easier to compare performance improvements on ECE and accuracy. Also, can the ranking variance be provided?
2. It might be better if there is standalone performances of the method without combining with temperature scaling? Without temperature scaling, does this method show improvement compared to other training-based calibration algorithms? It's fine if there's no improvement since the ultimate goal of this paper is calibratability. But if there's an improvement, it indicates the method is still of some value when there is no validation set for post-hoc calibration (it is true that we can also leave a validation set for post-hoc calibration, but it also involves trade-off between the gain from validation set and the gain integrating the validation set into training).
3. PLT still uses weight decay. Given the previous findings indicating that weight decay is quite sensitive in enhancing calibratability, how was the hyperparameter for weight decay chosen? Are hyperparameter selections needed?
4. In Table 2, it would be better to clearly indicate that these are the performances combined with temperature scaling.
5. The caption for Table 4 mentions different models (Table 4: The comparative results on Tiny-ImageNet with ResNet-18 (top) and ResNet-50 (bottom)), but the references seem to differentiate between training from scratch and fine-tuning. It seems inconsistent.
6. Some typos: figure 1 (h): there is a "!". Page 7 "Weight decay for frozen layers" line 2: there are two "same".

---

> ### Author Response · Authors · 2023-11-16
> **Response to Reviewer 7Hnm (Part 1)**
>
> ***
> Thanks for your questions and insightful comments. The following are our responses:
> ***
>
> ***Q1.*** *`The description of calibratability should be related to the final calibration performance after combining with a post-hoc calibration method?`*
>
> **A1.** We comprehend your perspective. While certain train-time methods also exhibit substantial reduction magnitudes after post-hoc calibration like SD in Figure 1, their final ECE is still higher than that of standard training. Thus, we measure calibratability by evaluating the final calibration performance after post-hoc calibration. In this paper, we specifically analyze the weight decay regularization method. It is observed that for models trained with strong weight decay, both the final calibration performance and the proportion of calibration performance reduction in the post-calibration phase are worse compared to standard training.
>
> ***
>
> ***Q2.*** *`It's interesting that figure 1 g/h have different tendencies. Do you have any intuition why different dataset have different tendency for ECE dynamics?`*
>
> **A2.** Considering the diverse difficulty levels of these datasets, each dataset may pose different challenges in terms of calibratability. Nevertheless, it is shown that the general comparison between strong regularization and standard training remains steady. Furthermore, we have introduced new experiments involving larger weight decay on CIFAR-10. In this result (please refer to this link:  https://anonymous.4open.science/r/anonymously-used-1-31E1/Calibrated%20ECE.png), we observe that as we increase the strength of weight decay, models on CIFAR-10 also experience a noticeable increase in calibrated ECE throughout training.
>
> ***
>
> ***Q3.*** *`Regarding that this compression process might compromise the model's calibratability", is it because the model, during the process of pushing each sample towards the class center, loses the uncertainty information of each sample in terms of their confidence?`*
>
> **A3.** Thanks for sharing this perspective. This perspective offers an intuitive and valuable insight into understanding the information loss in the model from the standpoint of feature distribution. However, we would like to suggest that not all the reduction of within-class variance will compromise the model's calibratability. As mentioned earlier, it is believed that all layers of a neural network undergo information compression gradually from the bottom to the top, but Figure 2 demonstrates that only the top few layers result in a degradation of calibratability. We suggest that only **over-compression** renders the model more challenging to differentiate between easy and hard samples and hence makes model less calibratable.
>
> ***
>
> ***Q4.*** *`What about binning-based methods or kernel-density-estimation-based post-hoc calibration methods?`*
>
> **A4.** We choose to compare calibratability based on scaling-based methods for two primary reasons: (1) Scaling-based methods, particularly temperature scaling, are widely utilized and proven to be effective post-hoc calibration techniques. (2) Scaling-based methods generates dense confidence distribution, which is advantageous for evaluation since they can circumvent compression in the post-hoc stage, alleviating concerns about an inappropriate comparison.
>
> ***
>
> ***Q5.*** *` I wonder if the reconstruction error is related to the dimension of the feature embedding of each layer.`*
>
> **A5.** Thans for the valuable question. It is widely believed that compression occurs continuously across all layers with the deepening of the network layers, regardless of how the relative parameter quantities vary across different layers, and it usually contributes to accuracy improvement. We want to underscore that, as discussed in Section 3.1, it is only the over-compression in the few top layers that adversely affects calibratability. In Figure 3, the crucial factor is the variance among different models rather than the specific values of reconstruction loss, and comparing different weight decay coefficients holds greater significance than comparing between layers.
>
> ***

---

> ### Author Response · Authors · 2023-11-16
> **Response to Reviewer 7Hnm (Part 2)**
>
> ***Q6.*** *`Regarding the organization and some typos (Experimental section Q1, Q4 and Q6).`*
>
> **A6.** Thanks for your feedback. We have made revisions to our paper based on the feedback from the reviewers, addressing aspects such as writing and organization.
>
> ***
>
> ***Q7.*** *`It might be better if there is standalone performance of the method without combining with temperature scaling? Without temperature scaling, does this method show improvement compared to other training-based calibration algorithms?`*
>
> **A7.** Thanks for the suggestion. We added the comparison of raw ECE between different methods in Table 6 in Appendix (you can also refer to this link: https://anonymous.4open.science/r/anonymously-used-2-F2DF/raw_ECE.jpg). The experimental results indicate a significant advantage in the average performance of our method in terms of raw ECE. As you mentioned, although train-time calibration is not our primary objective, this excellent performance makes our method more practical in specific scenarios. It is noteworthy that despite the fact that the comparative methods are all train-time calibration strategies, their performance is subpar, which aligns with the results in our Figure 1 (where their performance exhibits a V-shaped trend in terms of hyperparameters). This is due to their requirement for highly precise hyperparameter strategies to achieve good train-time calibration performance. By the way, we think these results further validate the inadvisability of pursuing train-time calibration.
>
> ***
>
> ***Q8.*** *`The caption for Table 4 mentions different models (Table 4: The comparative results on Tiny-ImageNet with ResNet-18 (top) and ResNet-50 (bottom)), but the references seem to differentiate between training from scratch and fine-tuning. It seems inconsistent.`*
>
> **A8.** Each subtable in Table 4 contains results both with and without the use of pre-trained parameters, as well as results for different input feature sizes. In the first and second columns of each subtable, you can observe the corresponding specific configurations.
>
> ***
>
> ***Q9.*** *`Given the previous findings indicating that weight decay is quite sensitive in enhancing calibratability, how was the hyperparameter for weight decay chosen? Are hyperparameter selections needed?`*
>
> **A9.** In our experiments, we applied the same weight decay coefficient (1e-4) to both PLT and standard training to ensure a fair comparison. In fact, when taking weight decay into account, there exists a trade-off between accuracy and calibrated ECE, while our method remarkably achieves this trade-off without the need to select specific hyperparameters.
>
> ***

---

> ### Author Response · Authors · 2023-11-23
>
> Dear Reviewer,
>
> We sincerely thank you again for your great efforts in reviewing this paper. We have gone through your points one-by-one and tried to address them carefully. Please don’t hesitate to let us know if you have any further questions and comments.
>
> Best regards,
>
> Authors

---

### Official Review · Reviewer_x8eC · 2023-10-31

**Soundness:** 1 poor
**Presentation:** 2 fair
**Contribution:** 3 good
**Rating:** 5
**Confidence:** 4

**Summary:**

This work investigates the calibratability and accuracy of various regularization techniques and a post-hoc calibration method. The author find a U shape calibration phenomenon where the calibration ability of the low layer and high layer representation is poor (high ECE), while the calibration ability of the middle layer representation is high (low ECE).
The author further proposes a progressively layer-peeled training method (PLT) which gradually freezes higher layers during training.

**Strengths:**

- This paper gives a good background study and related work review.
- The U-shape calibration ability phenomena in Figure 2 is interesting and intuitive.
- The proposed PLT method is also simple and easy-to-understand.

**Weaknesses:**

- This work points out the reason for poor calibration as strong compression. For example in section 1 *"to ensure that the top layers of the neural network do not excessively compress information, thereby enhancing the model’s calibratability"* and section 3.2 *"significantly compress the sample information, thereby reducing the model calibrability"*.  However, this is no evidence to support this point. For example, a post-hoc calibration method that changes the temperature of softmax can change calibration ability without any information compression.

- The explanation of experimental results doesn't align with the experiment itself. For example, in table 1 (weight decay = 1e-3), top layer (index 17) improves validation accuracy from 71.5 to 75.9. That is NOT a *"limited accuracy gain"*.  But this paper explains this result as *"We can observe that for all the weight decay policies, the top layers significantly improve the calibrated ECE with limited accuracy gain."*

- Wrong / unclear experiment settings. This paper claims that **applying weight decay to frozen layers is one key to the success of the proposed method** (in section 3.2). By my understanding, however, it is meaningless to apply weight decay to frozen layers. Because "frozen layers" mean the corresponding parameters are fixed.  How to apply weight decay on fixed parameters?

**Questions:**

- typo error "same same" in section 3.2.
- I suggest using "increase ...." instead of "improve the calibrated ECE" in section 3.1. Because "improve" means "make it better!"

---

> ### Author Response · Authors · 2023-11-16
> **Response to Reviewer x8eC**
>
> ***
> Thanks for your questions and insightful comments. The following are our responses:
> ***
>
> ***Q1.*** *`There is no evidence to support the claim that "significantly compress the sample information, thereby reducing the model calibrability" and “the top layers of the neural network do not excessively compress information, thereby enhancing the model’s calibratability". For example, a post-hoc calibration method that changes the temperature of softmax can change calibration ability without any information compression.`*
>
> **A1.** Thanks for the comment. After identifying the U-shape calibratability phenomenon as shown in Figure 2, we try to understand why this happens with experimental analyses in the rest parts of Section 3. As is shown in Figure 3, if you focus on the difference of models trained with strong and weak weight decay on these two metrics (rather than comparing different layers), you can clearly see the correlation between the information loss and calibratability in the top layers’ features. We believe that this observation is strong evidence for our claim. The experiments of weak classifier hypothesis, in which we consider that freezing top layers can avoid over-compression during training, have further confirmed this point.
>
> Regarding your mentioned post-hoc calibration example, we would like to clarify that we focus on the impact of training-time compression on calibratability in this paper, since we think this “calibration-unfriendly” over-compression occurs along with parameter updates during the training process, rather than in the post-calibration phase. More importantly, our perspective is more accurately stated as **over-compression** specifically deteriorates calibratability, rather than asserting that “any compression results in bad calibration” or “the change in calibration performance necessarily accompanies information compression”.
>
> ***
>
> ***Q2.*** *`In table 1 (weight decay = 1e-3), top layer (index 17) improves validation accuracy from 71.5 to 75.9. That is NOT a "limited accuracy gain". But this paper explains this result as "We can observe that for all the weight decay policies, the top layers significantly improve the calibrated ECE with limited accuracy gain."`*
>
> **A2.** Thanks for the comment. The statement can be reformulated as follows: “The overall results across different weight decay coefficients reveal a trend where the top layers notably enhance calibrated ECE with only marginal gains in accuracy”. In the specific scenario you highlighted, it is noteworthy that even with weight decay = 1e-3, considering the results for layer 15 and 17 **(accuracy increasing from 74.7 to 75.9 and ECE increasing from 2.86 to 5.03)**, the accuracy improvement is relatively modest compared to the increase in calibrated ECE.
>
> ***
>
> ***Q3.*** *`This paper claims that applying weight decay to frozen layers is one key to the success of the proposed method (in section 3.2). By my understanding, however, it is meaningless to apply weight decay to frozen layers. Because "frozen layers" mean the corresponding parameters are fixed. How to apply weight decay on fixed parameters?`*
>
> **A3.** Thanks for the comment. We would like to clarify that "freezing top layers" implies that we do not perform gradient descent on the corresponding parameters, as we believe that the calibration-unfriendly over-compression occurs along with parameter optimization process but not with simple weight decay. This can be easily achieved using the following pseudocode:
> >if layer_idx < cut_point:
> >
> >&nbsp;&nbsp;&nbsp;&nbsp;Perform SGD: $\theta_t \leftarrow \theta_{t-1}-\gamma (g_t+\lambda \theta_{t-1})$;
> >
> >else:
> >
> >&nbsp;&nbsp;&nbsp;&nbsp;Only perform wegith decay:  $\theta_t \leftarrow (1-\gamma \lambda) \theta_{t-1}$.
>
> As mentioned in the paper, we empirically found applying weight decay to the frozen top layers leads to favorable predictive performance. The following experimental results (accuracy) provide supporting evidence. We also added Figure 9 in Appendix to reveal these results. As our goal is to achieve good calibration without the loss on predictive performance, this operation is maintained in our main experiments.
> > | | &nbsp;&nbsp;SVHN&nbsp;&nbsp;|  &nbsp;&nbsp;CIFAR-10&nbsp;&nbsp;  |  &nbsp;&nbsp;CIFAR-100&nbsp;&nbsp;  |  &nbsp;&nbsp;Tiny-ImageNet&nbsp;&nbsp;  |
> > |  :----:              |    :----:   |     :----:   |     :----:   |     :----:  |
> > |  Keep WD|  **95.96**   |   **94.70**   |   **75.18**   |    **51.97**|
> > | Drop WD       |  94.46   |   94.19   |   72.63   |    50.77|
>
> ***

---

> > ### Author Response · Authors · 2023-11-23
> >
> > Dear Reviewer,
> >
> > We sincerely thank you again for your great efforts in reviewing this paper. We have gone through your points one-by-one and tried to address them carefully. Please don’t hesitate to let us know if you have any further questions and comments.
> >
> > Best regards,
> >
> > Authors

---

> ### Comment · Reviewer_x8eC · 2023-11-23
>
> Thanks for your answer!
>
> To A1: I get your point **"over-compression specifically deteriorates calibratability,"**. Please emphasize this point more.
>
> To A2: From pure number comparison, ECE 2.86 --> 5.03 seems to double the score (seems a lot) while accuracy 74.7 --> 75.9 only increases a little bit (1~2 percent). But we need to check the meaning of each score. (For example, one can easily apply log() or exp() on these scores to dramatically change the score range). In practice, 74.7 to 75.9 may not be a huge gain but 71.5 to 75.9 is never a "limited accuracy gain". There could be a better expression than "limited accuracy gain".
>
> To A3: Now I know the method. I suggest describing this method directly. **"freezing top layers"** has its own meaning, aka totally fixing parameters.
>
> I raise my score from 3 to 5, because now I feel it interesting with your answer.  I couldn't raise more because some parts of the paper are confusing, such as "freezing top layers" and "limited accuracy gain".

---

> > ### Author Response · Authors · 2023-11-23
> >
> > Thanks for your feedback!
> >
> > To **[To A1]**: Thanks for your suggestion. As we highlighted in our responses to R2, it is widely believed that all layers of a neural network undergo information compression gradually from the bottom to the top, but our experiments demonstrate that only the top few layers adversely affect calibratability. We will make this point clearer in the revised version.
> >
> > To **[To A2]**: Thanks for your comment, and we understand your perspective. By the way, do you think the statement **"The overall results across different weight decay coefficients reveal a trend where the top layers notably enhance calibrated ECE with only marginal gains in accuracy"** is preferable here?
> >
> > To **[To A3]**: Thanks for the valuable suggestion. I now understand that this description could lead to confusion, and we will clarify this in the revised version.
> >
> > Thank you again for your constructive feedback. It is greatly beneficial in improving our work!

---

### Author Response · Authors · 2023-11-22
**Thanks for your efforts in reviewing this paper**

Dear Reviewers,

We sincerely thank you again for your great efforts in reviewing this paper. We have gone through your points one-by-one and tried to address them carefully. Please don’t hesitate to let us know if you have any further questions.

Best regards,

Authors

---

### Meta-Review · Area_Chair_k3jg · 2023-12-07

**Metareview:**

The paper investigates the concept of calibratability, which aims at assessing how amenable a model is to be recalibrated in post-training phase. It presents empirical evidence that the overtraining of the top layers in neural networks poses a significant obstacle to calibration, while these layers typically offer minimal improvement to the discriminability of features. Given this observation, it introduces a weak classifier hypothesis that guides the development of a progressively layer-peeled training (PLT) method to to enhance model calibratability.

Reviewers have acknowledged that some of the key observations identified by the paper are interesting, which may potentially contribute to calibrated training of deep learning models. Meanwhile, they also pointed out that more concrete evidences on both the theoretical and empirical sides are still needed to justify the proposed approach. There are still confusions regarding the meaning of calibratability and the evaluation is not entirely convincing, either.

Multiple reviewers engaged a discussion with the authors during the rebuttal period. However, they were not totally convinced in the end and two reviewers maintained a negative rating. In order to make a more convincing case, substantial changes are needed to adequately address the important concerns as summarized about along with many detailed comments from individual reviewers.

**Justification For Why Not Higher Score:**

More concrete evidences on both the theoretical and empirical sides are still needed to justify the proposed approach.

**Justification For Why Not Lower Score:**

N/A

---

### Decision · Program_Chairs · 2024-01-16

Reject